# CD34 and CD105 Microvessels in Resected Bone Specimen May Implicate Wound Healing in MRONJ

**DOI:** 10.3390/ijerph182111362

**Published:** 2021-10-29

**Authors:** Antonia Marcianò, Antonio Ieni, Rodolfo Mauceri, Giacomo Oteri

**Affiliations:** 1Department of Clinical and Experimental Medicine, University of Messina, 98124 Messina, Italy; 2Department of Human Pathology of Adults and Developmental Age, Gaetano Barresi, University of Messina, 98124 Messina, Italy; antonio.ieni@unime.it; 3Department of Surgical, Oncological, and Oral Sciences, University of Palermo, 90127 Palermo, Italy; rodolfo.mauceri@unipa.it; 4Department of Biomedical, Postgraduate School of Oral Surgery, Dental Sciences and Morphofunctional Imaging, University of Messina, 98124 Messina, Italy; giacomo.oteri@unime.it

**Keywords:** medication-related osteonecrosis of the jaw, angiogenesis, surgical treatment, treatment outcome

## Abstract

Clinical treatment outcome of MRONJ (medication-related osteonecrosis of the jaw) surgery despite radical osseous removal and primary closure healing still shows differences in terms of outcome and disease recurrence. The study aims to assess the rate of angiogenesis of MRONJ lesions in order to understand the impact of angiogenesis and neoangiogenesis status on MRONJ surgical treatment outcome. This is the first study correlating microvessel density with prognosis in MRONJ surgically-treated patients. The immunohistochemical expression of CD34 and CD105 in MRONJ specimens obtained from surgically-treated patients was evaluated. The most vascularized areas detected by CD34 and CD105 were selected and the microvessel density value of the samples was registered. Samples were retrospectively divided according to the clinical outcome of MRONJ surgical treatment, dividing patients into two groups, “healed” and “not healed”. Statistical analysis was performed to assess if neovessels could influence treatment outcome in patients undergoing radical surgery. In the examined cohort, this value was highly predictive of better treatment outcome after radical surgery of MRONJ. Understanding of angiogenesis-dependent factors deserves further attention as a future target for MRONJ prevention and therapies.

## 1. Introduction

Medication-related osteonecrosis of the jaw (MRONJ) is “an area of exposed bone or bone that can be probed through an intra-oral or extra-oral fistula(e) in the maxillofacial region that has persisted for more than eight weeks, with current or previous treatment with anti-resorptive or anti-angiogenic agents and no history of radiation therapy to the jaws or obvious metastatic disease to the jaws” [1]. Drug-induced avascular injury is a cause for concern because of the limited therapeutic chances and there is an intellectual gap in our understanding of the pathogenesis of this disease, moreover this condition strongly influences the patient’s quality of life. Surgical treatment may offer benefits to MRONJ patients [2,3], and its indications have expanded over time from being limited to advanced stage [4,5] to being considered more effective when performed in early stage [6,7,8,9]. Nevertheless, surgery has shown different success rates in the literature, depending on different variables which have an impact on the surgical intervention outcome [10].

Sometimes, subsequent debridements are necessary to obtain clinical healing. Successful outcome has been associated with complete removal of all affected tissue to avoid the progression of MRONJ and minimize reintervention, thus radical intent is always aimed at first surgery.

Evaluation of the factors affecting the necessity for revision surgery in MRONJ should be taken into consideration to decide on the most appropriate surgical technique.

Indeed, recurrence-related factors of MRONJ are not fully elucidated [11,12,13,14,15]. Several investigations have suggested that the underlying disease, the duration of administration of MRONJ-related medications, the presence of bacterial infections as well as the adopted treatment strategies act as factors in MRONJ recurrence [11,12,13]. Patients affected by osteoporosis are more prone to heal in comparison to cancer patients [13,14]. It appears that this could be due to the concomitant administration of anti-cancer medications which may play a role in the occurrence of adversity [15], although a clinical and symptomatological remission could still be experienced by cancer patients regardless of the underlying malignancy [16]. Concomitant treatment with corticosteroids or tobacco smoking are reported to be other individual factors that can inhibit the bone healing process in MRONJ patients [17]. A further relevant parameter that favors a positive outcome of surgery could be the event triggering the outbreak of MRONJ [14,15]. MRONJ localization can also affect the outcome of surgery; indeed, proximity to the maxillary sinus can lead to interfering correlations with the condition of chronic sinusitis which would be detrimental to healing [14]. A crucial aspect is considered the presence of a focal lesion with margins clearly detected by the use of a TC of maxillary bones [18]. The adopted surgical technique is reported to be one of the most relevant factors that affects the outcome of the MRONJ treatment [15], according to the extent of the surgical procedures and the closure technique, which strongly influences treatment outcome [11,12,19,20,21]. Furthermore, it has been also reported that some intrinsic characteristics of the MRONJ lesion may result in a better or worse outcome of the procedure. Mature bacterial biofilms are now identified as potential critical triggers in the pathogenesis of drug-related osteonecrosis of the jaw, which can also have a negative influence on disease resolution, as well as on its onset [22]. Thus, the perioperative antibiotic regimen has a significant influence on the chances of disease recurrence [23,24].

In the clinical experience of the University of Messina (Italy), despite radical surgery and targeted antiobiotic therapy, MRONJ outcome still shows differences in terms of outcome and disease recurrence [25,26].

Assuming that the rate of neoangiogenesis in resected bone specimen may be implicated in wound healing after MRONJ surgery, this investigation was designed to determine the prevalence of microvessels in MRONJ biopsies and to examine the relationship with the associated perioperative outcomes.

Starting from this assumption, the immunohistochemical expression of two very reliable markers—CD34 and CD105, effective for the study of angiogenesis and neoangiogenesis—was evaluated in MRONJ specimens obtained from surgically-treated patients in order to understand the impact of the angiogenesis status on MRONJ treatment outcome.

CD34 and CD105 (also known as Endoglin) are endothelial antigens that have been chosen due to their recognized role as direct markers of the degree of vascularization and neoangiogenesis. Specifically, CD105 is a cell membrane glycoprotein related to newly formed blood vessels, while CD34 is expressed in both mature and newly formed vessels [27,28].

Considering all the above-mentioned variables, the secondary objective of this study has been to investigate the different conventional factors affecting outcome and their impact on angiogenesis in the cohort of surgically-treated patients of the University of Messina.

## 2. Materials and Methods

### 2.1. Study Design

The study was conducted at the Department of Human Pathology of Adult and Developmental Age of the University of Messina and coordinated from the Department of Biomedical, Dental and Morphofunctional Imaging Sciences of the University of Messina. The surgical specimens were obtained from surgically-treated MRONJ patients at the Center for Treatment of the Osteonecrosis of the Jaws (University of Messina, Italy). Biopsies were taken from the bone tissue including the necrotic area with a rim of adjacent bone [29,30]. The necrotic tissue itself was excluded from the analysis [31]. Samples were retrospectively divided according to the clinical outcome of MRONJ surgical treatment, dividing patients into two groups, “healed” and “not healed”. “Healing” was defined as clinical wound healing without dehiscence or evidence of recurrence [9,32]. Minimum follow-up was established in at least 6 months.

### 2.2. Inclusion Criteria

MRONJ diagnosis had to be performed according to the definition of the Italian Societies of Oral Medicine and Maxillofacial Surgery (the SICMF-SIPMO staging system) [18,33]. Only patients who underwent radical surgery were considered eligible for the study. In addition, patients needed to have a biopsy sample sent to the pathology lab for processing at the time of surgery.

### 2.3. Surgery

Patients are referred to the Osteonecrosis of the Jaw Treatment Center, School of Dentistry, University of Messina mostly by their oncologist. On arriving at the center, patients are diagnosed with MRONJ based on the clinical and radiological findings in order to distinguish focal and diffused forms. Routine procedures at first examination include oral swab and pharmacological treatment prescription with systemic antibiotics. Initial treatment is amoxicillin plus clavulanic acid in combination with metronidazole at 250 mg; subsequently, patients are switched to targeted antibiotic therapy on the basis of the antibiogram result. Eight to ten weeks after the initiation of pharmacological treatment, unchanged and progressive forms urdergo surgical treatment [34]. The systematic application of this work flow ensures homogeneity in the patient sample in terms of antibiotic therapy (empiric vs. targeted therapy) and time to intervention (defined as time from MRONJ diagnosis to surgical procedure). The surgical approaches were defined according to literature [9,10,17,23,35,36] as previously described by our study group [25,26]. The surgery was performed in loco-regional anesthesia with intra-oral approach and consisted in the resection of the necrotic bone until reaching bleeding vital bone. Access to the osteonecrotic lesion was provided by a mucoperiosteal flap with total thickness sufficiently large to include the margins of the necrotic bone. The affected bone can be removed using ultrasonic bone surgery device. After removing the necrotic segment, if a sufficient quantity of soft tissue to obtain closure by first intention is present, the vestibular and the lingual mucoperiostal flaps were directly sutured on the defect without any release incision; otherwise, closure could be obtained using mucosal advancement flaps to allow for a tension-free suture. Figure 1, Figure 2, Figure 3, Figure 4, Figure 5 and Figure 6 show pre-surgical assessment and specimen harvesting during surgical procedure (Figure 1, Figure 2, Figure 3, Figure 4, Figure 5 and Figure 6).

### 2.4. Study Variables

Patient characteristics were analyzed. Investigated variables were patient related and treatment related. Demographic data (age and gender), primary disease (cancer or osteoporosis), type of medication (zoledronic acid, denosumab or oral bisphosphonates), duration of antiresorptive treatment (referring to osteoporotic vs. cancer patients and expressed in months) were reported and analyzed. MRONJ clinical features (localization, stage of MRONJ) were registered and analyzed.

### 2.5. Immunohistochemical Analysis

Histomorphometrical analyses were performed in a blinded fashion without knowledge of the clinical features and treatment outcome of the patients corresponding to individual biopsies [29]. Four micrometer-thick consecutive sections were cut from the paraffin blocks and submitted to the immunohistochemical procedures against CD105 and CD34. For the CD105 epitope retrieval, specimens were treated with proteinase K (S3020, DAKO Cytomation) at room temperature for 15 min, while CD34 antigen was unmasked by microwave oven pre-treatment in 10 mM, pH 6.0 sodium citrate buffer for 3 cycles × 5 min. Then slides were incubated at 4 °C overnight with the primary monoclonal antibodies against CD105 (DAKO Corporation, Denmark, clone SN6 h, w.d. 1:50) and CD34 (DAKO Corporation, Denmark, clone QBEnd10, w.d. 1:50); a sheep anti-rabbit immunoglobulin antiserum (Behring Institute; w.d. 1:25) was used and the bound primary antibody was visualized by avidin–biotin–peroxidase revealing by the Vectastain Rabbit/Mouse Elite Kit, according to the manufacturer’s recommendations. To reveal the immunostaining, the sections were incubated in darkness for 10 min with 3–3′ diaminobenzidine tetra hydrochloride (Sigma Chemical Co., St. Louis, MO, USA), in the amount of 100 mg in 200 mL 0.03% hydrogen peroxide in phosphate-buffered saline solution (PBS). Nuclear counterstaining was performed by Mayer’s haemalum. Specificity of the procedure was confirmed by omitting the primary antiserum or changing it with normal rabbit serum/phosphate buffered saline solution (PBS pH 7.4). In addition, samples of human placenta were applied as a positive control for CD105. The quantification of microvessels was performed according to the procedure elsewhere described [37]. Necrotic areas were excluded. In detail, the three most vascularized areas detected by CD105 were firstly selected (hot spots) under 40× field. Microvessels were then counted in each of these areas under a 400× field. Single endothelial cells or cluster of endothelial cells, with or without a lumen, were considered to be individual vessels. The mean value of three ×400 field (0.30 mm^2^) counts was verified as the microvessel density (MVD) of the slide. Successively, the MVD value was converted into the mean number of microvessels/mm^2^ for statistical investigations. The vessels were evaluated using a Nikon microscope by two independent observers blinded to the clinico-pathological data. The same procedure was carried out for CD34 expression on equivalent slides. Figure 7 and Figure 8 are explanatory of the performed immuno-staining procedure. All the remaining images have been uploaded as electronic Appendix A for reader consultation (See Appendix A for supporting content).

### 2.6. Statistical Analysis

Statistical analysis was performed to assess if neovessels could influence treatment outcome in patients undergoing radical surgery. Data are presented as means ± standard deviation (SD). Student’s *t* test for means and Fisher’s exact test (*p* < 0.05) for the other values for low numbers were used to compare means between two different groups. A *p*-value below 0.05 was considered statistically significant.

## 3. Results

Among the MRONJ cases referred to the Center for Treatment of the Osteonecrosis of the Jaws (University of Messina, Messina, Italy), 15 patients fulfilled the above-mentioned inclusion criteria.

### 3.1. Conventional Risk Factors

The characteristics of patients together with the conventional risk factors for MRONJ recurrence are reported in Table 1.

Nine patients were enrolled in the “healed” group and 6 patients in the “not healed” group. No statistically significant difference between the “healed” (68.22 ± 8.45) vs. the “not healed” (69.17 ± 6.05) group was registered in relation to mean age of the patients (*p*-value = 0.8166). The majority of the patients in the “healed” group were women (*n* = 7; 77.8%) with the remaining (*n* = 2; 22.2%) as male patients. In the “not healed” group, 1 (16.7%) subject was a female patient and 5 (83.3%) were male patients. The sex of the subjects represented a variable that significantly influenced post-surgical healing (*p* = 0.0406). In relation to primary disease, in the “healed” group there were 6 cancer patients (66.7%) and 3 patients affected by osteoporosis (33.3%), while in the “not healed” group all the patients had cancer. The underlying pathology had not a statistical significant impact on healing (*p* = 0.2286). Among patients enrolled in the “healed” group, 3 patients were treated with zoledronic acid, 3 patients received denosumab and 3 patients were exposed to oral bisphosphonates, while in the “not healed” group almost all the patients received zoledronic acid (*n* = 5; 83.33%) with denosumab being administered to the remainig patient (*n* = 1). The type of antiresorptive medication was not significanty related to healing since no difference in the use of zoledronic acid (*p* = 0.1189) or denosumab (*p* = 0.6044) has been observed. In relation to the duration of therapy, this information has been stratified on the basis of the clinical indication of the administered medication (cancer vs. osteoporosis) and expressed in months. Cancer patients in the “healed” group received anti-resorptive treatment (35.33 ± 18.79) for a shorter period of time than cancer patients in the “not healed” group (40.33 ± 20.71). Median duration of anti-resorptive treatment in patients affected by osteoporosis was calculated only for the “healed” group (72.22 ± 51.73 months) as all the patients healed. Duration of antiresorptive therapy was not statistically related to a better outcome of surgical treatment (*p* = 0.6356). The most common location of MRONJ was the mandible in both “healed” (*n* = 7) and “not healed” (*n* = 3) groups, with the upper jaw being affected in *n* = 2 patients in the “healed” and 1 patient in the “not healed” group. In two cases of the “not healed” group, both jaws were affected. MRONJ location was not significantly related to post-surgical healing in lower (*p* = 0.3287) or upper jaw (*p* = 1.0000). According to the SIPMO classification, the most frequent stage of MRONJ in the “healed” group was stage IIa (*n* = 4; 44.4%) followed by stage IIb (*n* = 3; 33.3%). The remaining 2 patients were stage Ia (*n* = 1) and IIIb (*n* = 1). In the “not healed” group, there was a higher proportion of stage IIIb (*n* = 3; 50%) patients followed by stage IIb (*n* = 2; 33.3%) and IIa (*n* = 1; 16.67%). In this case, series MRONJ stage did not affect surgical treatment outcome.

### 3.2. CD34 and CD105 Expression

Fifteen MRONJ tissue samples were evaluated. Table 2 shows the immunohistochemical expression of the angiogenetic factors evaluated.

Angiogenesis as expressed by the median CD34 rate was higher in the “healed” group (MVD = 66.57) compared with “not healed” patients (MVD = 3.94). This difference in angiogenesis was significant between the two groups (*p* = 0.015756). Newly formed angiogenesis-related capillaries, which stained positively for CD105, were detected only in the specimens of “healed” patients (MDV = 19.33). The inhibition of neoangiogenesis was strongly significantly related to surgical treatment outcome (*p* = 0.000973).

### 3.3. Correlation between CD34 and CD105 Expression and Conventional Risk Factors

The possible relationship between patient’s characteristics and angiogenesis is examined in Table 3.

Pertaining to the correlation between angiogenetic biomarkers and conventional risk factors in the observed samples, sex was significantly related to the presence of CD34 stained capillaries (*p* = 0.0256) although it was shown to be uninfluential to the expression of CD105 (*p* = 0.1319). Primary disease didn’t affect vascularization (*p* = 0.5165) nor neoangiogenesis (*p* = 1.0000). The different administration of zoledronic acid in the “healed” (*p* = 0.5692) vs. the “not healed” (*p* = 0.3147) as well as the assumption of denosumab in the “healed” (*p* = 1.0000) vs. the “not healed” (*p* = 0.5692) groups was not statistically significant. The duration of the anti-resorptive therapy itself appeared to be unrelated to the vascularization of the jaw bone and the neovessels formation (*p* = 0.6084). In relation to MRONJ location, the upper jaw was significantly related to a greater angiogenesis (*p* = 0.0769). In this analysis, the MRONJ lesions divided in the three SIPMO stages didn’t show a statistical difference in the local expression of the investigated angiogenetic biomarkers.

Table 4 summarizes the anti-cancer therapies undertaken in order to explore the concomitant use of antiangiogenic agents that can affect the investigated parameters.

## 4. Discussion

The response of clinically-similar MRONJ lesions to the same surgical treatment may be vastly different. Therefore, to improve clinical care and to give an optimal treatment, recurrence-related factors must be identified.

The aim of the present study was to investigate the role of altered angiogenesis and its potential relationship with wound healing in MRONJ surgery starting from the consideration that angiogenesis is a critical component of MRONJ development and may be implicated in wound healing [31]. Indeed, the hypothesis that the impairment/inhibition of angiogenesis has an important role in the development and maintenance of MRONJ seems to be the most relevant pathogenetic theory explaining the pathway in which necrosis occurs [38,39]. It involves avascular necrosis through VEGF and PECAM-1 suppression confirming the interplay between angiogenesis and osteogenesis in bone integrity maintenance [40,41]. The few studies published on the topic have mostly been restricted to the comparison between MRONJ vs. health subjects [42]. According to Gao et al., the suppression of angiogenesis and osteogenesis (identified by a reduction in CD31 in MRONJ models) could be significant in the potential mechanism of MRONJ itself [43]. The results by Wehrhan et al. showed that the capillary area related to CD31-associated vascularity was slightly reduced in the MRONJ-related specimens compared to normal mucoperiosteal tissue, likewise the newly-formed angiogenesis-related capillaries which were positively stained for CD105, were detected to a lesser extent in MRONJ-related specimens than in mucoperiosteal tissue not exposed to bisphosphonates [31]. Compromised angiogenesis would most likely be involved in post-surgical healing, although other aspects of the vasculature (e.g., blood flow) could contribute to MRONJ [44,45]. The results of the present study showed a greater number of newly-formed angiogenesis-related capillaries detected by CD105 staining in tissue belonging to healed MRONJ patients (*p* = 0.000973) and a higher expression of CD34 (*p* = 0.015756), indicating that inhibition of angiogenesis could be implicated in MRONJ healing.

Patients with non-healable wounds are expected to have a lower microvessel density associated with lower success rate than those who have a higher microvessel density and a major healing potential.

Microvessel count, reflecting the angiogenesis, appears to be deserving of further study. Further prospective studies are necessary to investigate these findings systematically; nevertheless, a significant trend emerged in the presented results and could be of interest.

The present study also addresses the conventional factors affecting MRONJ recurrence and their correlation with the immunohistochemical parameters expression. There are several variables that have a statistically significant impact on MRONJ post-surgical healing: gender, patients’ underlying pathologies, anti-resorptive drugs used, time to treatment, MRONJ stage and others [11,12,23].

Almost none of the conventional factors taken into consideration (age, type of medication, duration of antiresorptive therapy, underlying pathology, MRONJ location and MRONJ stage) were significantly related to MRONJ surgical treatment outcome with the exception of sex. This data is considered to be due to the prevalence of women in the examined cohort (68%) reflecting the fact that MRONJ-related medications are mostly prescribed for post-menopausal osteoporosis and breast cancer. Although in this analysis the underlying pathology and the type of antiresorptive medication had not a statistical significance, it is noteworthy that all patients with osteoporosis assuming oral bisphosphonates completely healed after surgery. This is consistent with Martins et al., who observed significant differences in outcomes and time to healing according to primary disease (*p* < 0.05) [46].

Data from the literature confirm the importance of primary disease on the outcome of the therapy; indeed, cancer patients treated with high-potency bisphosphonates may experience a poor treatment outcome in comparison to denosumab-related MRONJ, likewise patients with prolonged bone anti-resorptive therapy may show a worse outcome [13,14].

In relation to MRONJ location, the upper jaw was significantly related to a greater angiogenesis (*p* = 0.0769) but this didn’t translate into a better surgical outcome. This may be due to the anatomical differences between the maxillary and mandibular bones and owing to the intimate anatomical relationship with the sinus [47,48,49]. Patients attributable to the three stages of the disease were treated, with a higher frequency of cases attributable to stage II. It has been reported that early stage treated with radical surgery showed a better treatment outcome [50]. Nevertheless, the MRONJ stage did not play a significant role in the incidence of surgical failure (*p* = 0.2352), in this case series. Analysis of the correlation between neoangiogenetic biomarkers and conventional risk factors showed that the presence of CD34 stained capillaries was significantly correlated with sex and MRONJ location. MVD as assessed by CD34 and CD105 expression was significantly associated to surgical treatment outcome. Comparison was very difficult to perform because of the limited size of the study samples. Nevertheless, within this group of selected patients we found that some markers of angiogenesis were useful tools to characterize patient’s surgical healing and were associated with disease outcome. However, consideration should be given to concomitant anti-cancer therapies (Table 4) since the use of antiangiogenic agents can affect the investigated parameters, contributing to angiogenesis inhibition, microenviroment alterations and immune response [51,52,53,54]. Our results highlight the need for a wider and more reliable investigation on neoangiogenesis biomarkers in MRONJ.

There are certain limitations to our study. For one, we had limitation in sample size, which may have limited the disclosure of statistically significant relationships between MVD and MRONJ surgical treatment outcomes. We had enrolled all patients who underwent surgery for MRONJ in our Department during a four-year period starting from the year 2016; however, recruitment experienced a severe decline during the COVID-19 pandemic, now slowly returning to normal volumes. We chose to recruit only patients treated with radical intent to avoid bias resulting from different prognoses of palliative treatment. Moreover, according to the SICMF-SIPMO diagnostic criterion for MRONJ are clinical and radiological [34,35], thus the clinical routine of our Center biopsy of MRONJ specimens is not performed systematically. Therefore, during the study period, a total of 77 MRONJ patients were enrolled and a total of 34 surgical procedures were performed with radical intent. Among these, 15 patients met the inclusion criteria. However, it must also be said that single-center studies related to the surgical treatment of MRONJ always present limited data and the cohorts presented in literature are generally small. Anyhow, it must be recognized that the small sample size limited the interpretation of any significant clinical impacts; for this reason, future development of the research aims to expand the patient cohort with progressive patient recruitment.

At the time of writing, follow-ups range from a minimum of 6 months to a maximum of 6 years; when expanding our cohort, it is our intention to take in consideration not only failures but recurrences also which may occur in a different location. Nevertheless, despite the limited sample, we were able to demonstrate a significant impact of the expression of CD34 and CD105 on MRONJ outcome.

## 5. Conclusions

This is the first study correlating MVD with prognosis in MRONJ surgically-treated patients. In the examined cohort, the MDV as assessed by CD34 and the presence of neovessels by CD105 was highly predictive of better treatment outcome after radical surgery of MRONJ. Exploring CD34 and CD105 expression in MRONJ surgically-treated patients’ specimens revealed that the healing potential of patients could be also influenced by the presence of neovessels in the MRONJ lesion. To identify diagnostic markers of drug-induced vascular injury would add value in MRONJ risk management and prognosis [55,56]. The implication on clinical practice of having such information available would be important to schedule surgery, post-operative wound management and follow-up examinations on the basis of the success/failure risk. Furthermore, since tooth extraction is the major precipitating event in MRONJ, investigating the role of altered angiogenesis in wound healing may have future implications in MRONJ prevention as well as in the surgical treatment. Finally, we believe that the understanding of angiogenesis-dependent factors deserves further attention as a future target for MRONJ therapies. Indeed, on the basis of these results, a potential protection of the jaw from the negative influence of zoledronic acid and/or denosumab by locally enhancing angiogenesis could be imagined for MRONJ prevention and treatment.

Our research group is presently investigating the expression of several angiogenetic markers on a wider sample of MRONJ patients and controls to provide a better definition of their potential clinical value; and in the field of the biotechnology, we aim at the development of new treatment strategy adjuvants or alternatives to surgical treatment for all non-eligible patients for whom only palliative therapies are currently available.

## Figures and Tables

**Figure 1 ijerph-18-11362-f001:**
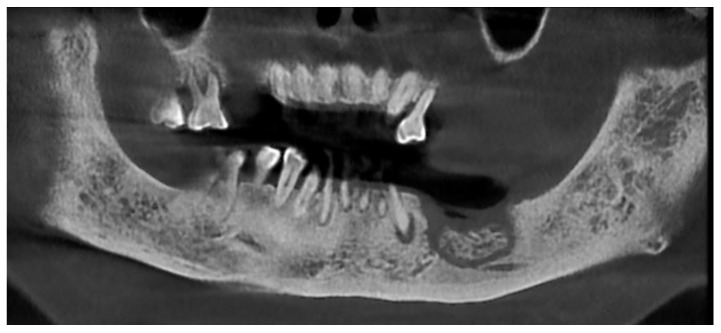
Radiologic appearance of focal MRONJ lesion in left mandible.

**Figure 2 ijerph-18-11362-f002:**
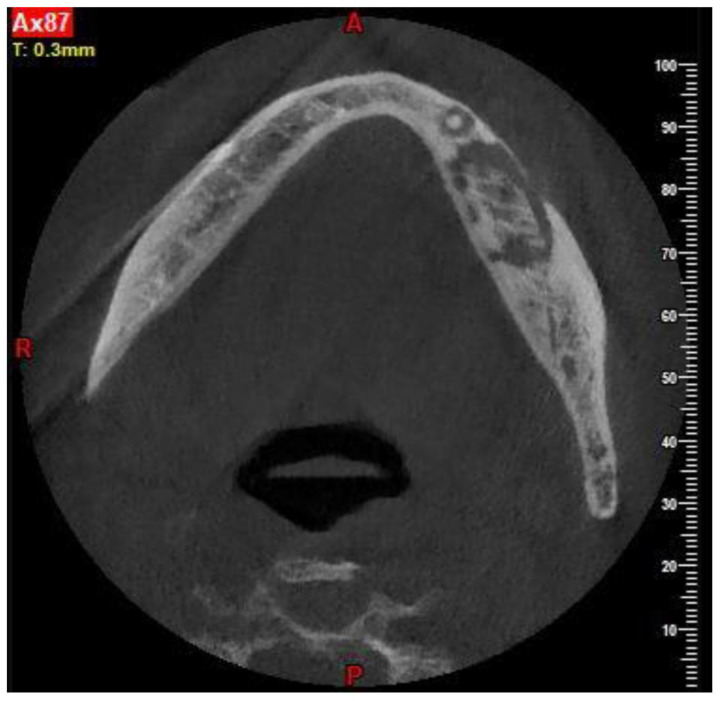
CBCT showing the necrotic bone extension.

**Figure 3 ijerph-18-11362-f003:**
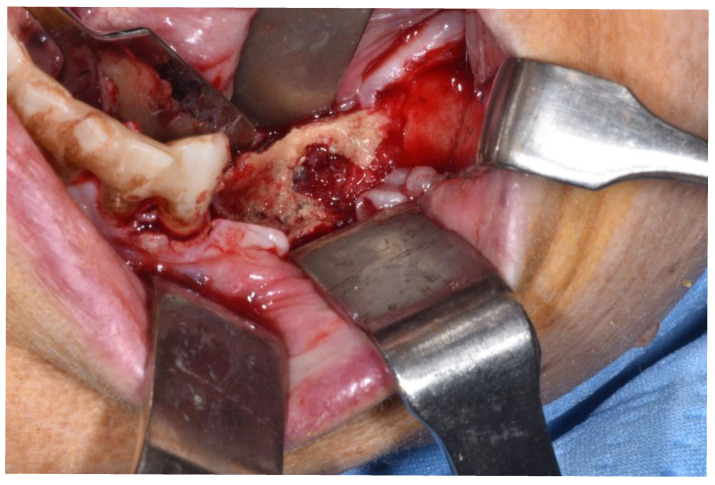
Clinical appearance when elevating mucoperiosteal flap.

**Figure 4 ijerph-18-11362-f004:**
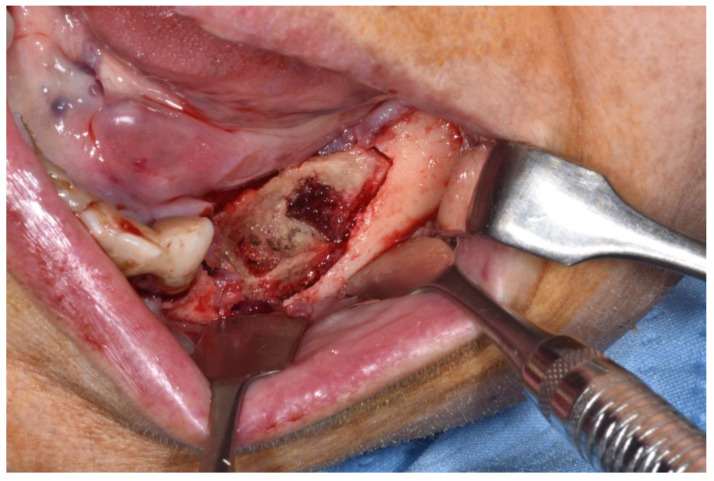
Sub-marginal resection performed using piezoelectric device.

**Figure 5 ijerph-18-11362-f005:**
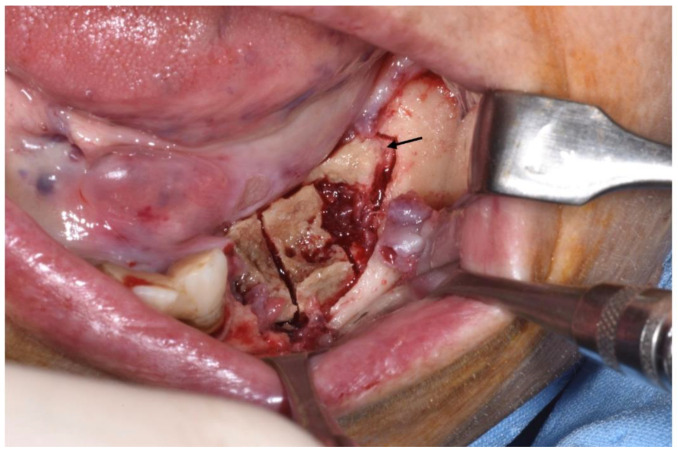
Intraoperative aspect. Necrotic bone segment was observed macroscopically and cut, dividing it into three parts: a transitional zone (segment located in the mesial third of the sequestrum), totally necrotic tissue (the middle part of the segment) and healthy adjacent tissue (the distal area of the segment). The arrow schematically indicates the portion used in this case.

**Figure 6 ijerph-18-11362-f006:**
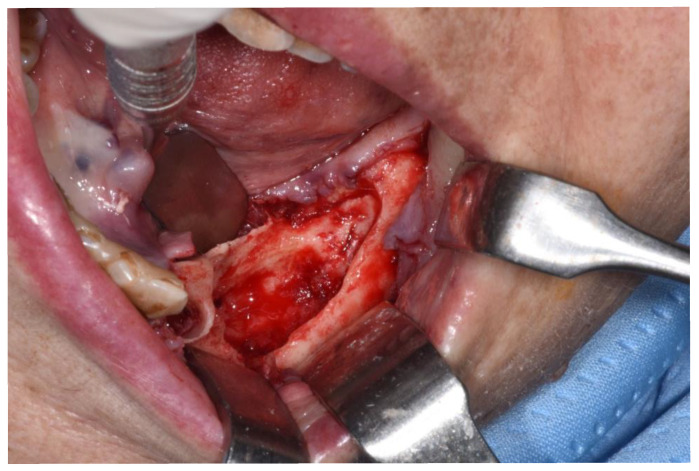
Revision of the cavity until observing bleeding bone.

**Figure 7 ijerph-18-11362-f007:**
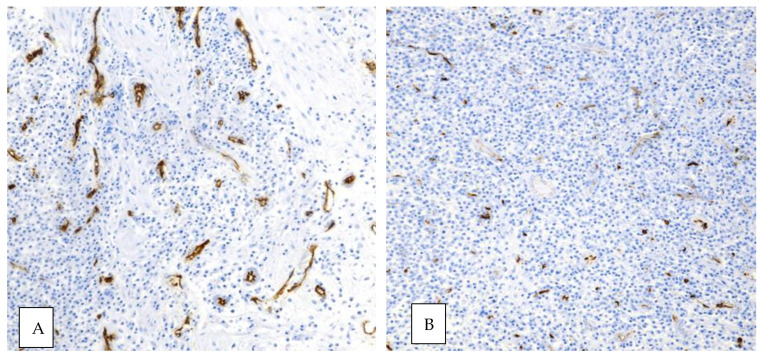
The immunopositive staining pattern of CD34 ((**A**) original magnification ×200, nuclear Mayer’s haemalum counterstain) and CD105 ((**B**) original magnification ×20, nuclear Mayer’s haemalum counterstain) in the same area.

**Figure 8 ijerph-18-11362-f008:**
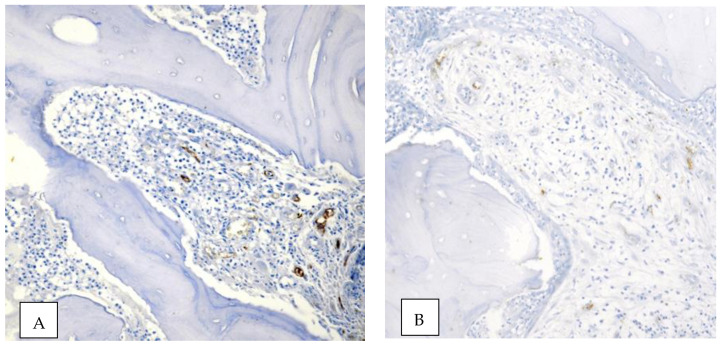
A lower immunostaining pattern of CD34 ((**A**) original magnification ×200, nuclear Mayer’s haemalum counterstain) and CD105 ((**B**) original magnification ×20, nuclear Mayer’s haemalum counterstain) in a non-healed patient.

**Table 1 ijerph-18-11362-t001:** Study population features and conventional risk factors for MRONJ recurrence after surgical treatment, divided according to clinical outcome.

	Healed (n° 9)	Not Healed (n° 6)	*p*-Value *
Age years (average)	68.22 ± 8.45	69.17 ± 6.05	0.8166
Sex (%)			
Female	7 (77.8)	1 (16.7)	**0.0406**
Male	2 (22.2)	5 (83.3)	
Primary disease (%)			
Cancer	6 (66.7)	6	0.2286
Osteoporotic	3 (33.3)	/	
Antiresorptive medications (%)			
Zoledronic acid	3	5 (83.33)	0.1189
Denosumab	3	1	0.6044
Oral bisphosphonate	3	/	
Duration of antiresorptive therapy in months (SD)			
Cancer patients	35.33 ± 18.79	40.33 ± 20.71	0.6356
Osteoporotic patients	72.33 ± 51.73	/	
MRONJ Location (%)			
Lower jaw	7	3	0.3287
Upper jaw	2	1	1.0000
Both jaws	/	2	
MRONJ Stage (%)			
I a	1	/	
I b	/	/	
II a	4 (44.4)	1	0.5804
II b	3 (33.3)	2	1.0000
III a	/	/	
III b	1	3	0.2352

MRONJ: medication-related osteonecrosis of the jaw. SD: standard deviation. * Student’s *t* test for means and Fisher’s exact test (*p* < 0.5) for the other values for low numbers.

**Table 2 ijerph-18-11362-t002:** Immunohistochemical parameters in patients with MRONJ divided according to treatment outcome.

	Healed (n° 9)	Not Healed (n° 6)	*t*-Value	*p*-Value
CD34 MVD (v/mm^2^)—median rate	66.57	3.94	−2.33203	0.015756
CD105 MVD (v/mm^2^)—median rate	19.33	0	−3.59139	0.000973

MVD = microvessel density.

**Table 3 ijerph-18-11362-t003:** Correlation between angiogenetic biomarkers and conventional risk factors.

	CD34		*p*-Value *	CD105		*p*-Value *
	Present N = 11	Absent N = 4		Present N = 8	Absent N = 7	
Sex (%)						
Female	8	/	**0.0256**	6	2	0.1319
Male	3	4		2	5	
Primary disease (%)						
Cancer	8	4	0.5165	6	6	1.0000
Osteoporotic	3	/		2	1	
Antiresorptive medications (%)						
Zoledronic acid	5	3	0.5692	3	5	0.3147
Denosumab	3	1	1.0000	3	1	0.5692
Oral bisphosphonate	3	/		2	1	1.0000
Duration of antiresorptive therapy in months						
≤24 months	5	1	0.6044	4	2	0.6084
>24 months	6	3		4	5	
MRONJ Location (%)						
Lower jaw	2	1	1.0000	1	2	0.5692
Upper jaw	9	1	**0.0769**	7	3	0.1189
Both jaws	/	2		/	2	
MRONJ Stage (%)						
I a	1	/		1	/	
I b	/	/		/	/	
II a	5	/		3	2	1.0000
II b	3	2	0.5604	3	2	1.0000
III a	/	/		/	/	
III b	2	2	0.5165	1	3	0.2821

MRONJ: medication-related osteonecrosis of the jaw. SD: standard deviation. * Student’s *t* test for means and Fisher’s exact test (*p* < 0.5) for the other values for low numbers.

**Table 4 ijerph-18-11362-t004:** Summary of anti-cancer treatments of the 15 MRONJ cancer patients.

Patient Number	Age/Gender	Site of Carcinoma	Cancer Medications
1	70/male	Prostate	Degarelix
2	68/male	Myeloma	Bortezomib + Melphalan; Daratumumab; Lenalidomide + Dexamethasone
3	79/male	Prostate	Enantone + 22RaCl
4	72/male	Myeloma	Lenalidomide
5	63/female	Breast	Fulvestrant Palbociclib Letrozolo
6	63/male	Renal cell	Sunitinib; Everolimus; Sorafenib
7	60/male	Prostate	Bicalutamide; Abiraterone acetate
8	59/female	Breast	Tamoxifen, Letrozole, Lapatinib, Fulvestrant, Exemestane, Docetaxel, Capecitabine, Vinorelbine, Cyclophosphamide, Methotrexate, Eribulin, Doxorubicin, Palbociclib + Fulvestrant, Fluorouracil (5-FU)
9	67/female	Breast	Everolimus; Paclitaxel; Exemestane; Doxorubicin; Anastrozole; Eribulin
10	72/male	Breast	Cyclophosphamide + Methotrexate
11	80/female	Breast	Everolimus + Exemestane
12	79/female	Myeloma	Lenalidomide + Dexamethasone; Bortezomib + Melphalan + Prednisolone

## Data Availability

Anonymized data of all patients are archived in a database specifically created with the purpose of conducting this research.

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
