# Peer review of "CD34 and CD105 Microvessels in Resected Bone Specimen May Implicate Wound Healing in MRONJ"

_ijerph, 2021, doi:10.3390/ijerph182111362_

Round 1

Reviewer 1 Report

The authors presented an interesting manuscript illustrating the prognostic role of CD34- and CD105-stained microvessels on the healing outcomes of MRONJ. Although the pathological role of angiogenesis inhibition is still not clarified, this manuscript demonstrated the potential clinical indications of angiogenesis for treatment outcomes. The originality is high and the article is clearly organized. However, the small sample size limited the interpretation of any significant clinical impacts. The statistical methods are not correctly used for the exploration of a prognostic factor. Moreover, the "radical surgery" should be detailed. The 6-month follow-up is also questionable for MRONJ, which may recur a few years later even heal at 6 months. The authors should also include figures on the IHC results and surgical operation.

Author Response

We sincerely thank the reviewers for their thoughtful advices since we believe the suggested changes are an important improvement for our work and were highlighted in the text in green color.

The "radical surgery" should be detailed.

We have welcomed in a special way the reviewer's suggestion and we have better detailed the whole Materials and Methods section by adding the mossing informations to the paragraph describing the “Surgery”.

The surgery was performed in loco-regional anesthesia with intra-oral approach and consisted in the resection of the necrotic bone until reaching bleeding vital bone. Access to the osteonecrotic lesion was provided by a mucoperiosteal flap with total thickness sufficiently large to include the margins of the necrotic bone. The affected bone can be removed using ultrasonic bone surgery device. After removing the necrotic segment if a sufficient quantity of soft tissue to obtain closure by first intention is present, the vestibular and the lingual mucoperiostal flaps were directly sutured on the defect without any release incision otherwise closure could be obtained using mucosal advancement flaps to allow for a tension-free suture. Figures 1 to 6 show pre-surgical assessment and specimen harvesting during surgical procedure (Fig. 1-6).

The 6-month follow-up is also questionable for MRONJ, which may recur a few years later even heal at 6 months.

Success of surgical treatment was defined in the Materials and Methods section as wound healing without dehiscence, without clinical signs of infection or signs of failed bone healing. The clinical result was evaluated based on the findings recorded at least at 6 months but follow-up ranged from a minimum of 6 months to a maximum of 6 years. 

To better explain this concept, in the discussion section we added a sentence to underline that a recurrence of MRONJ could be situated in a site different from the previous localization and that it our intention to expand the cohort taking in consideration also these forms.

The authors should also include figures on the IHC results and surgical operation.

Figures (7-8) which are explanatory of the performed immuno-staining procedure has been added as suggested. All the remaining images has been uploaded as electronic supplemental material for reader consultation (Appendix A).

Reviewer 2 Report

General comment

" Prognostic Impact of CD34 and CD105 Expression and Neo-vessels after Radical Surgery for MRONJ Treatment "

It is very interesting to to assess the rate of angiogenesis of MRONJ lesions in order to understand the impact of vascularization and neoangiogenesis status on MRONJ surgical treatment outcome. However, there are a few corrections that are essential to meet the standard for publication. Please refer to the following comments.

1) This study is an important issue for dental and oral surgery and is very interesting. But unfortunately your study design is ineligible. Most importantly, the target patients have many different baselines. Comparisons between configured groups that are too biased are inappropriate. Please reconsider the patient division.

2) The authors also point out that the number of patients is small as a limitation of this study. But this limitation could be resolved. Because MRONJ is not a rare disease. It can be solved by extending the research period. Please reconsider.

3) Please show the actual image of immunostaining. Since there are few cases in particular, it is better to attach all images as supplementary materials. Readers are interested in the differences from case to case.

Author Response

We sincerely thank the reviewers for their thoughtful advices since we believe the suggested changes are an important improvement for our work and were highlighted in the text in green color.

Study design is ineligible. Most importantly, the target patients have many different baselines. Comparisons between configured groups that are too biased are inappropriate. Please reconsider the patient division.

As we described in the Methods section patients were considered eligible for the study when they had a diagnosis of MRONJ and underwent radical surgical treatment, to make it more clear we also added that patients needed to have a biopsy sample sent to the pathology lab for processing at the time of surgery.

We thank the reviewer for the suggestion nevertheless we would like to try to keep the two groups “healed” and “not healed” without further patients stratification in an effort to give the reader a clear vision of the treatment-related outcome. 

Furthermore it seemed more interesting to diversify the cohort to test the technique on a heterogeneous population of both dysmetabolic and cancer patients.

Modifying them would certainly be appropriate but we fear we will not be able to convey our vision of the different characteristics affecting disease progression/improvement. 

Making the groups homogeneous (only cancer patients or only osteodysmetabolic patients) could eventually prevent the identification of the characteristic that affect the expression of the angiogenesis status which in turn impacts the outcome.

Because MRONJ is not a rare disease. It can be solved by extending the research period. Please reconsider.

In reply to the reviewer’s comment: study period begins with the first surgically treated patient in 2015 and ends in 2021. The retrospective nature of the study limited us to patients for whom biopsy specimens were available. In the experience of the University of Messina as Referral Center in Eastern Sicily, Italy the trend in hospitalizations has been fairly constant, with about 3-4 admissions on average per month. Furthermore biopsy is not a MRONJ diagnostic criterion thus we didn’t use to perform it systematically, but we intend to expand the cohort as suggested in the future and have already begun patient recruitment and we added a sentence in the discussion section to better detail it. 

Please show the actual image of immuno-staining. Since there are few cases in particular, it is better to attach all images as supplementary materials. Readers are interested in the differences from case to case.

Thanks for the suggestion. As advised Fig.7-8. which are explanatory of the performed immuno-staining procedure has been added to the main text while all the remaining images (n. 30) has been uploaded as electronic supplemental material for reader consultation (Appendix A).

Round 2

Reviewer 1 Report

The authors revised the manuscript as suggested. The added figures enhance the readability. But severe flaws remain as I commented last time——the small sample size limited the interpretation of any significant clinical impacts. The statistical methods are not correctly used for the exploration of a prognostic factor.

As I wrote to the editors last time, the authors may consider rewriting this manuscript as a small pilot study, especially not to overstate the clinical implications.

Some specific comments are in below.

1. the Title may be revised to not overstate the "prognostic impact", because the limited sample size and statistics do not support to do a good prognostic paper. The authors may consider the title as: "CD34 and CD105 microvessels in resected bone specimen may implicate primary wound healing in MRONJ"

2. the authors are suggested to carefully give the definitions "angiogenesis", "vascularization", "neoangiogenesis", "microvessel".

3. please do not overstate "predictive" or "prognosis", because most data are "associative", and the small sample size actually only provides a "descriptive" profile

4. i still not capture the "radical surgery", the attached case is more "sequestretome" or "marginal resection". I noticed some staged III patients were included. Are there any radical "bone resection" or "bone reconstruction" performed? like the fibular flap.

5. for neovessel IHC, usually the endothelial and pericytes are stained. the authors should explain more why they choose CD34 and CD105, the provided Ref#27 #28 do not give information on this.

6. there is still questionable for the role of vascularization in MRONJ, the authors should discuss why CD34 and CD105 may implicate primary wound healing.

7. please provide a schematic graph illustrating which portion of bone specimen was taken for IHC, which portion of necrotic bone was excluded. Should I understand as the marginal portion of resected bone, or the "marginal microvessel density" is indicative for primary wound healing?

8. the ethical approval for human subjects was not stated in the context?

9. the authors evaluated the microscropic microvessels by counting the number of positive stains, so, the exact numbers of positive stains in Figure 7B and Figure 8b look quite similar, although the size of positive stains in Figure 7B are bigger. Figure 8A should be changed with equivalent bone marrow space as in other figures.

10. there is no illustration on Table 4.

11. How was the treatment strategy for those non-healed patients?

12. "vascolarization" should be "vascularization"

13. The introduction and discussion should be re-organized.

Author Response

We thank the reviewer for her/his thorough evaluation and constructive recommendations for improving this manuscript. 

The reviewer is correctly stating that the small sample size limited the interpretation of any significant clinical impacts. 

However, in our work we report preliminary data that nevertheless seemed worthy of mention. Results will be implemented over time but a significant trend still emerges. We however understand the reviewers concern thus following his/her comment and the other reviewers' considerations we softened the tones considering this manuscript as a small pilot study. 

In response to reviewer specific comments:

  1. We edited the title as suggested using "CD34 and CD105 microvessels in resected bone specimen may implicate primary wound healing in MRONJ” 

2. We thank the reviewer for pointing out the difference between the terms. We replaced the word     vascularization with angiogenesis and differentiated only angiogenesis and neoangiogenesis. 

3. We revised the text paying more attention to the choice of terms

4. This is indeed a pertinent question. Although we have a standard surgical technique for surgical interventions which include incision, mucoperiosteal flap elevation, debridement of the necrotic bone segment  both through decortication or through sub-marginal resection and primary intention closure we always have to radical intent since it has been reported to be more effective. A fibula flap reconstruction has never been performed in our cases

5 We have welcomed in a special way the reviewer's suggestion to discuss on the use of pericytes markers. As suggested by the literature these should be CD31 NG2 and PDGFR-beta. In our investigation we chosed CD34 CD105 instead since these are very reliable markers effective for the study of neoangiogenesis as extensively described in the literature

6. We agree with the reviewer when he/she said the role of vascularization in MRONJ is a field to be explored. We strenghtened in the text that it is not the markers themselves but the neoangiogenesis quantified with these two markers that is involved in healing

7. As suggested we have proceeded to implement readability and clarity of figure 5 which has been modified by adding an arrow. The arrow schematically indicates the portion used in this case.  In all specimens it was always analyzed the mesial or distal portion of the segment excluding the central area. Pertaining the relationship between marginal microvessel density ans primary wound healing as correctly inferred by the reviewer this it is not indicative

8. The ethical approval has not been reported in the materials and methods since it was stated in the “Patents” section as requested by the guidelines for the Authors

9. Figure 8A has been changed with its equivalent as requested

10. Table 4 has a title. We didn’t add a legend at the foot of the table because all terms are written without abbreviations

11. Not healed patients are usually put in medical treatment with antiseptics/antibiotics. Sometimes subsequent debridements are necessary. See also Zirk M, Kreppel M, Buller J, Pristup J, Peters F, Dreiseidler T, Zinser M, Zöller JE. The impact of surgical intervention and antibiotics on MRONJ stage II and III - Retrospective study. J Craniomaxillofac Surg. 2017 Aug;45(8):1183-1189

12. This has been modified according to point 2

13. As suggested by the reviewer, we re-organized the “Introduction” and “Discussion” sections using more cautious words and stressing the concept of radical intent at first intervention and the need to remove all affected tissue to avoid the progression of MRONJ and minimize reintervention

We believe the suggested changes are an important improvement for our work and were highlighted in the text in green color.

Reviewer 2 Report

Thanks for the many manuscript corrections.

However, further revisions to your manuscript are needed for publication.

Your study included only biopsy cases. If you have the opportunity to receive treatment three or four times a month, there should be a large number of cases without specimens.

Please add the breakdown and results of those cases, as well as the percentage of cases for which samples were collected.

This is essential to improve the transparency of the study.

Author Response

We thank the reviewer for her/his thorough evaluation and constructive dialogue. 

We agree with him/her when correctly stating that the small sample size limit the interpretation of any significant clinical impact of the results. 

We added the required data pertaining patients enrollment in the Discussion section. A total of 77 MRONJ patients were enrolled during the study period, 2016-2019 + 2 years of pandemic, and a total of 34 surgical procedures were performed with radical intent excluding palliative interventions from the list. Among these 15 patients met the inclusion criteria.

Unfortunately recruitment that had initiated before the pandemic in the last two years has been temporarily suspended due to Covid-19. Despite this outbreak enrollment continues to slowly increase up to the usual numbers.

We understand the reviewers concern however in our work we report preliminary data that seemed worthy of mention. Results will be implemented over time but a significant trend still emerges and could be of interest.

Following the reviewer’s comments and the other considerations from reviewer #1 we softened the tones considering this manuscript as a small pilot study.

In the years to come we hope to be able to recruit a greater number of patients treated with the same surgical technique and with routine biopsy.

Pertaining the outcome of non-biopsy cases our group already presented the results of others larger observational studies about MRONJ surgical treatment and we expect to reach also in this case the same number in a few years.

Changes were highlighted in the text in green color.